# Recent Development of Mechanical Stimuli Detectable Sensors, Their Future, and Challenges: A Review

**DOI:** 10.3390/s23094300

**Published:** 2023-04-26

**Authors:** Shushuai Zhu, Dana Kim, Changyoon Jeong

**Affiliations:** School of Mechanical Engineering, Yeungnam University, 280 Daehak-Ro, Gyeongsan 38541, Republic of Korea

**Keywords:** pressure sensor, single-force detection, normal force detection, shear force detection, strain detection, multiforce testing, piezoresistive, capacitive

## Abstract

By virtue of their wide applications in transportation, healthcare, smart home, and security, development of sensors detecting mechanical stimuli, which are many force types (pressure, shear, bending, tensile, and flexure) is an attractive research direction for promoting the advancement of science and technology. Sensing capabilities of various force types based on structural design, which combine unique structure and materials, have emerged as a highly promising field due to their various industrial applications in wearable devices, artificial skin, and Internet of Things (IoT). In this review, we focus on various sensors detecting one or two mechanical stimuli and their structure, materials, and applications. In addition, for multiforce sensing, sensing mechanism are discussed regarding responses in external stimuli such as piezoresistive, piezoelectric, and capacitance phenomena. Lastly, the prospects and challenges of sensors for multiforce sensing are discussed and summarized, along with research that has emerged.

## 1. Introduction

Sensing of various force types, such as pressure [1,2,3,4,5], shear [4,6,7,8], strain [9,10,11,12], torsion [2,3,13,14], and bending [2,3,15,16], is a fundamental building block of a multitude of advanced applications, including wearable devices [1,2,17,18,19,20,21,22,23,24,25,26,27], human–computer interactions (HMI) [28,29,30,31,32,33], automatic industries, and automatic driving [34,35]. With the increasing demand for these industries, there is dramatic need for devices that sense various force types in one sensor. However, current sensors are still in an early stage of development and require further innovation in terms of sensor structure and materials for multiforce sensing in one sensor. Key technologies of this sensor for detecting various forces and distinguishing them include electric signal analysis, design of sensor, and material and flexible forms.

Figure 1 shows sensors that monitor and detect a single force [36,37,38,39,40] and multiforces [41,42,43,44], and illustrates their applications [45,46,47]. For application in many industries, these sensors require the sensing capabilities of multiforces and multiforce sensing and distinguishing them for accurately required force calculations. Most developed sensors can detect a single force, and sensors detecting multiforces have definite a limitation when many forces act on the sensor simultaneously. In particular, various physical quantity sensing and discriminating are important factors for many industries where various physical quantities are applied simultaneously. Therefore, sensing various physical forces has attracted growing interest, and many studies have been reported. Structures that enable sensing of various physical forces have been developed and combined with electrical layer for sensing performance. Sensors that sense only a single force show a very simple structure and easy fabrication process. However, for detecting various types of forces, structure and materials of sensors are complex and the fabrication process is very complex for embodying these sensors. In addition, fabricated sensors with highly complex structure and materials for multiforce sensing have an obvious limitation in the detection of multiforces simultaneously and cannot distinguish them perfectly. Electric signals from a sensor are the only signals that can perceive an external force and it is difficult to distinguish each external force. A multilayered sensor [48,49] was developed to overcome this limitation, and several trials for sensing various forces were conducted. Despite this, the mass production of sensors is limited by the many electrodes required for sensing and the complicated fabrication process [50,51,52,53]. This review explains and summarizes the development of a sensor that can detect a single force and multiforce and its applications. The sensing mechanisms of sensors for various forces and their structure for sensing performance are also explained. 

## 2. Single-Force Detection

Pressure is one of the most measurable quantities and process parameters. Over the last few decades, scientists have researched the detection of pressure. With the development of technology, pressure sensors have emerged, including single-force detection and multiple-force detection pressure sensors. On the other hand, although multiple-force detection sensors can detect a force individually, other factors can influence the sensor, which is problematic when precision detection is required. The development of single-force sensors is also necessary because of the complex internal structure required to achieve multiple-force detection, which places stringent requirements on the working environment and increases manufacturing costs. Single-force sensors are used mainly for routine pressure detection, shear force detection, and strain detection. 

### 2.1. Normal Force Detection

For the detection of a single pressure, there are several main methods: piezoresistive, capacitive, field-effect transistor, and piezoelectric. Each method has its advantages and disadvantages. For example, piezoresistive sensors are simple to manufacture and have good linearity and acceptable sensitivity, but temperature influences these sensors and interferes with physical force sensing [54,55]. Capacitive pressure sensors allow high sensitivity and lower temperature hysteresis, but the linearity is poor [56,57]. Field-effect transistors can only detect very small pressures, whereas inductive models have good linearity but are less resistant to interference [58]. Piezoelectric models are self-powered but can only measure dynamic signals and are more affected by temperature [59,60]. Piezoresistive pressure sensors are used widely in single-force detection, particularly in recent years with the development of nanomaterials. Hybrid nanocomposites of exfoliated graphite nanoplatelet (xGnp) and multiwalled carbon nanotubes (MWCNT) were added to polydimethylsiloxane (PDMS) to fabricate sensors capable of detecting pressure (Changyoon Jeong et al. [61]). Mechanical and electrical tests were carried out to investigate the effects of nanofiller weight percentage, xGnP size, and the xGnP:MWCNT ratio on the compression modulus and sensitivity of the sensor. Figure 2a shows the principle of operation of the sensor and the effect of different xGnP:MWCNT ratios on the sensor. A pressure sensor was proposed based on porous polyurethane and graphene foam (Chunfang Feng et al. [62]). The principle of operation of this sensor is based on the contact of the conductive network leading to a change in resistance, as shown in Figure 2b. When subjected to pressure, the skeleton in the polymeric foam changes, leading to a change in the distance between the graphene sheets inside and a change in the conductive pathway of the sensor. Hence, the response can be the correspondence between pressure and resistance.

Organic field effect transistors (OFETs) have been used successfully as active components in display panels and sensor devices [63]. The OFET architecture can be ideal for active pixel elements in pressure sensor arrays on flexible substrates. An effective and convenient strategy was demonstrated using a simple solution treatment in which the thin poly(methyl methacrylate) (PMMA) of a controlled thickness can be combined with a thick polyelectrolyte of poly(acrylic acid) (PAA) to produce a new double layer dielectric, as shown in Figure 2c, which maintains high capacitance while greatly reducing the leakage current (Zhigang Yin et al. [64]). A flexible suspended-gate organic thin-film transistor (SGOTFT) was reported as a model platform for ultrasensitive pressure detection (Yaping Zang et al. [65]). The unique device geometry of SGOTFTs allows fine-tuning of the sensitivity using a suspended gate, successfully achieving low detection limit pressures of <0.5 Pa and short response times of 10 ms. As shown in Figure 2d, in SGMOSFETs, the gate is deformed under stress, which causes the capacitance of the dielectric layer to vary with the applied pressure, resulting in a pressure-dependent source leakage current at a constant gate voltage, the magnitude of which is known from the pressure. This sensor can be used for acoustic and pulse detection owing to its high sensitivity and ultralow power consumption.

Because of the unique mode of operation of capacitors, the two main factors affecting the change in capacitance are the distance between the poles and the squared area of the poles. Hence, capacitive sensors often detect normal force and shear, often changing only one factor when used as a single-force detection sensor [66]. A capacitive pressure sensor with a graded porous PDMS composite dielectric layer was fabricated by combining particle and emulsion templates (Juwon Hwang et al. [67]). When an external force was applied to the sensor, the spacing between the two substrates changed easily because of the high compressibility of the porous PDMS, as shown in the schematic diagram of the sensor in Figure 2e; H3 (graded PDMS with a 3:1 volume ratio of sugar–PDMS porous PDMS composite) has the highest sensitivity.

A piezoelectric pressure sensor based on a P(VDF−TrFE)/multiwalled carbon nanotube (MWCNT) composite was fabricated, as shown in Figure 2f, which is self-powered, portable, and safer than other types of pressure sensors (Aachen Wang et al. [68]).

**Figure 2 sensors-23-04300-f002:**
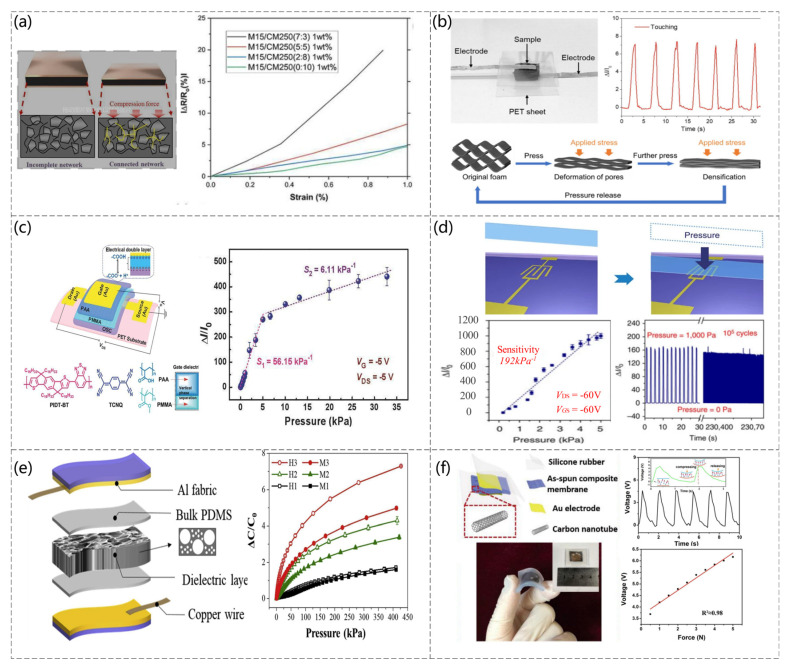
(**a**) Compressive sensing mechanisms in composites and normalized changes in resistance and compressive strain for different xGnP:MWCNT ratios. Reproduced with permission from ACS (2020) [61]. (**b**) Schematic diagram and mechanistic description of the sensor structure, finger touch test, and the change in the PU/G foam structure with stress. Reproduced with permission from Composites Part B (2020) [62]. (**c**) Schematic structure of the flexible OFET device and molecular structures of PIDT−BT, TCNQ, PAA, and PMMADS. Reproduced with permission from Advanced Science (2018) [64]. (**d**) Pressure sensing process, pressure response of source-drain currents at constant voltages V DS = −60 V and V GS = −60 V, and endurance test of SGOTFT at 1000 Pa. Reproduced with permission from Nat Commun (2015) [65]. (**e**) Diagram of the sensor structure and the change in capacitance for different sugar-to-volume ratios of applied pressure. Reproduced with permission from Composites Part B (2021) [67]. (**f**) Schematic representation of the sensor structure and flexibility and the output voltage response to an external force of 2.5 N at a frequency of 0.6 Hz. The inset shows the operating mechanism of the sensor and the linearity of the voltage generated versus the compression force in the range of 0.5 to 5.0 N. Reproduced with permission from Nanomaterials (2018) [68].

### 2.2. Shear Force Detection

The detection of shear forces has always been an important issue. There is a need for thin, wearable shear sensors in rehabilitation engineering, e.g., shear force detection between a prosthetic wearer and a prosthesis [69] or the measurement of a shear force between the sensor and the fluid interface in contact [70]. A shear force feedback system is used in optical microscopy [71]. There are generally two methods for accurate shear detection: piezoresistive and capacitive. Other forces, such as normal forces, should be avoided in detecting shear. A sheet shear force capacitive pressure sensor and its measurement system were developed (Shigeru Toyama et al. [69]). When a shear force is applied to the sensor surface, the electrode-patterned films move towards each other, causing the distance between the electrodes inside the sensor to change, resulting in a change in the capacitance of the sensor. The sensor also exhibits insensitivity to temperature changes. Figure 3a shows the structure of the sensor and the change in capacitance of the four capacitors when shear forces in different directions are applied. An upright piezoresistive cantilever with an embedded elastic material is proposed to detect a shear force applied to a surface, as shown in Figure 3b on the left side of the sensor operating principle and the right side of the graph corresponding to the change in shear force versus resistance (Kentaro Noda et al. [72]). A bionic, thin, and flexible liquid-metal-filled resistive PDMS microchannel shear-sensing skin was developed by observing the phenomenon of compression on one side and stretching on the other when a human finger was subjected to a shear force (Jianzhu Yin et al. [73]). Figure 3c shows the construction of the sensor, and the graph shows a linear relationship between the sensor response and the applied shear force for a constant normal force of 1 N. A piezoresistive shear detection sensor was also produced, where the distance between the two conductive sensor components depends on the shear force acting and the internal pressure (A. Chaykina et al. [74]). The magnitude and direction of the shear force are determined electrically by measuring the internal pressure at opening or closing. The structure of the sensor and the relationship between the shear force and the corresponding voltage are shown in Figure 3d.

### 2.3. Strain Detection

Strain can be observed using smart wearable devices that require the manufacture of strain sensors that can be stretched and measured accurately [75]. Several strain sensors have been proposed for making highly sensitive strain sensors using carbon nanomaterials on stretchable elastic substrates. For example, embedded 3D printing (e−3DP) was reported for the fabrication of strain sensors using carbon-based resistive inks embedded in a highly scalable elastic matrix using deposition nozzles, as shown in Figure 4a (Joseph T. Muth et al. [76]). The highly aggregated carbon nanotube (CNT) network embedded in the thermoplastic polyurethane (TPU) compound filament layer changes the CNT conductive network attached to the surface when the sensor is subjected to tensile forces, reflecting the correspondence between tensile deformation and sensor resistance. The sensor can have a strain of up to 400% (Qingqing Fan et al. [77]). Figure 4b presents a scanning electron microscopy (SEM) image of the CNT attached to the TPU surface and the sensor resistance versus pressure for different CNT ratios. Nanowires can vary the film resistance as the number of nanowires changes under pressure. For example, Morteza Amjadi et al. [78] used a sandwich structure to add a silver nanowire (AgNW) film between PDMS elastomers to detect the strain force. The working principle of the sensor is that the AgNWs are disconnected when they are stretched or bent (Figure 4c). The conductivity of the sensor also changes with the increase in the topology of the network. The experimental data agree with the simulated data. Xu Xiao et al. [79] reported a strain sensor based on a ZnO nanowire/polystyrene nanofiber hybrid structure on a PDMS film. The sensor could be strained by 50%, as shown in Figure 4d. The fabrication steps and relative changes in deformation rate and electrical resistance were examined. The excellent linearity of the sensor was noted (Figure 4d). In addition, there are strain sensors based on nanoparticles (NPs). Recoverable cracks are formed, resulting in changes in the current path, and the corresponding relationship between tensile force and current can be obtained. For example, a strain sensor based on Ag NP films was proposed. This sensor can measure tensile and compressive forces (Jaehwan Lee et al. [80]). Figure 4e shows the principle of the sensor and the change in tensile force and relative resistance. Shuai Wang et al. [81] reported on the fabrication method of a simple, low-cost, and scalable strain sensor. The strain sensor, based on the crack network mechanism, consists of multilayer carbon nanotubes (CNTs) films/polydimethylsiloxane (PDMS) composites. When subjected to strain, the sensor with a crack network can detect strain from propagated cracks, which can lead to changes in the conductive path and consequently in the resistance value. The sensitivity and sensing range of the sensor can be adjusted by precisely controlling the thickness of the multilayer CNT. The sensor was tested to have a wide sensing range of 100% and can detect strains down to 0.007%, as well as having a stability of 1500 cycles and a gauge factor of up to 87. Figure 4f shows the SEM image of the sensor and relationship between the strain pressure and resistance change after the composition of the sensor.

## 3. Multiforce Testing

In recent years, with the development of technology, new requirements have emerged for physical pressure sensors, particularly the emergence of intelligent robots, “haptic sensors”, “E-skin”, and “wearable devices”. There has been great development, and to simulate the natural sense of touch, tactile sensors are needed to make corresponding physical signals according to the stimuli of the environment to obtain relevant information from the physical interaction. On the other hand, developing pressure sensors that can simulate human perception for complex operations requires an ability to measure the normal contact force on the sensor surface, and the normal shear, shear angle, and bending torsion rotation are equally important. Pressure sensors, such as piezoresistive and capacitive sensors, that are highly sensitive, highly perceptive, and can measure a wide range of forces, can be developed in many ways.

### 3.1. Piezoresistive

Piezoresistive sensing converts mechanical displacements into electrical signals and can be varied to achieve high sensitivity and flexibility by changing the shape and composition of the sensor. Over the last few decades, researchers have worked on the metal piezoresistive effect [82,83] from the very beginning to the semiconductor piezoresistive effect [84,85,86,87], and later to the now-widely-studied piezoresistive effects in polymer composites [88,89,90,91]. Conductive polymers are popular among researchers because of their excellent stability, flexibility, mechanical properties, and electrical conductivity [92,93,94,95]. Therefore, they are often used as highly sensitive sensors. A hexagonal micropillar array was coated with ultrasonically sprayed single-walled carbon nanotubes to realize a biomimetic high-sensitivity mechanosensor (Changyoon Jeong et al. [96]). The sensor is based on PDMS micro-hexagonal column patterns fabricated by ultrasonically spraying single-walled carbon nanotubes (SWCNTs) on hexagonal columns, as shown in Figure 5a, which features an interlocking optimized geometry shape. Because the hexagonal micro-pillar array shows very high sensitivity, its contact area changes according to the action of different forces and can detect normal shear and bending forces. The PDMS is used as the electrode layer, and high-aspect-ratio ZnO NRs are grown vertically on the PDMS surface to achieve an interlocking structure (Min-Sheng Suen et al. [97]). The sensor can detect pressure, bending, torsional forces, and ambient temperature, as shown in Figure 5b. The resistance of the contact area changes when pressure is applied to the sensor. The high aspect ratio of the ZnO NRs is a crucial factor that affects the sensitivity of the sensor because of the increased contact area of the e-skin in the unit area. The design of a piezoresistive interlocking microdome array of sensors was inspired by the interlocking epidermal–dermal layers in human skin (Jonghwa Park et al. [98]). Similar to the stress concentration function of interlocking epidermal–dermal ridges, the stresses applied are amplified and concentrated to increase the sensitivity of the sensor. This sensor senses normal, shear, tensile, bending, and torsional forces depending on the stress type and method, as shown in Figure 5c. Chunhong Mu et al. [99] produced a highly sensitive graphene oxide (GO)/PDMS pressure sensor (Figure 5d). The sensor can detect normal and shear forces with specific reverses. When a pressure is applied to the sensor, certain compression of the GO/PDMS porous sublayer is induced, which would result in a decrease in resistance caused by the approach and overlap of the GO sheets in the GO/PDMS porous structure. The sensor is also extremely sensitive and can detect pressure under 25 Pa.

Guangming Cai et al. [100] reduced graphene oxide (RGO) in the presence of elastic nylon/PU fabrics, which serves as the skeleton for the RGO conductive layer, as shown in Figure 6a. When different forces are applied to the sensor, the RGO layer comes into contact with the adjacent yarn or fiber, resulting in a change in resistance. Therefore, the sensor can detect bending as well as torsional forces. The sensor is fixed to the finger and is used to detect minor strain caused by the bending of the finger. Therefore, it can be used as a potential application for wearable devices.

Cheng-Wen Ma et al. [101] patterned bucky paper (CNT films exhibiting a paper-like morphology). As a sensing element, a novel bucky paper patterning technique was proposed that achieved anisotropic sensing capabilities, flexibility, and ease of manufacture at low cost (Figure 6b). The carbon nanotube molecules in the bucky paper are interconnected and form a conductive network. When pressure is applied to the bump, the PDMS substrate deforms, resulting in a change in the resistance of the bucky paper sensing element. The prepared sensor can sense normal and shear forces and exhibits excellent sensitivity and repeatability.

A sensor for freestanding 3D polyaniline foam with a microcracking structure was presented (Shaodi Zheng et al. [102]). The sensor was based on polyaniline (PANI) foam transferred to a PDMS substrate, with microcracks induced in the foam skeletons following the fracture mechanics. When the sensor was subjected to a force, the width of the crack changed, resulting in a change in the resistance of the sensor (Figure 6c). Different mechanical stimuli produced different resistance changes. According to the experiment, the sensor could detect a normal pressure, shear force, and torsion.

A flexible composite sensor for normal force and shear detection was proposed, as shown in Figure 6d (Changyoon Jeong et al. [103]). The sensor was made by mixing CNTs and PDMS and casting them in a cylindrical mold. When a force was applied to the sensor, it changed the CNT network in the columnar structure and resistance.

A graphene-based composite fiber sensor with a “compression spring” structure was fabricated (Yin Cheng et al. [104]). This sensor can sense tensile strain, bending, and torsion, as shown in Figure 6e. The graphene fibers were also used as torque sensors, taking advantage of the inherent microstructure of polyester (PE) fibers. These fibers can differentiate the torsion direction according to the increase or decrease in resistance.

### 3.2. Capacitive

Compared to other pressure sensors manufactured on other principles, capacitive pressure sensors have attracted widespread attention owing to their simple structure, high sensitivity, temperature independence, wide monitoring range, good dynamic response, strong anti-interference capability, and low power [105,106,107,108,109,110,111,112,113,114]. The sensitivity of a sensor depends primarily on the insulating dielectric layer used because it is always embedded between the upper and lower conductive electrodes [115,116]. The deformation of the insulating dielectric layer causes a change in the capacitance of the sensor. In addition, the capacitance changes when the area between the two substrates and the area between them is changed, so researchers often change the composition or structure of the sensor to enable it to detect normal forces, shear forces, bending states, bending angles, and torsional forces. This paper reports a three-axis fully integrated differential capacitive tactile sensor surface-mountable on a bus line (Sho Asano et al. [117]). The sensor integrates a flip-bonded complementary metal–oxide semiconductor (CMOS) with capacitive sensing circuits on a low-temperature cofired ceramic (LTCC) interposer with Au through vias by Au–Au thermocompression bonding. The CMOS circuit and bonding pads on the sensor backside were electrically connected through Au bumps and the LTCC interposer, and an Au sealing frame formed the differential capacitive gap. As shown in Figure 7a, the magnitude and type of force were detected by the change in resistance when a normal force or shear force was applied. A capacitive sensing array consisting of PDMS structure and an flexible printed circuit board (FPCB) is presented, which is capable of detecting normal and shear forces and can be implemented easily using micromachining and FPCB technology, greatly improving the manufacturability of the sensor (Ming-Yuan Cheng et al. [118]). As with other capacitive sensors, the sensor is also based on the change in spacing between the two substrates for different types of force, thus determining the magnitude and type of force. Figure 7b shows the operating principle of the sensor and the change in capacitance of the sensor when subjected to shear forces and normal pressure in different directions. Daehwan Choi et al. [119] designed a pyramid-plug structure with high sensitivity to improve the sensitivity of the sensor, as shown in Figure 7c. The device was composed of pyramid-patterned ionic gel (an ionic and visco-poroelastic biocompatible elastomer). The ionic gel is an ionic thermoplastic polyurethane (i−TPU) composed of a polymer matrix (TPU) and ionic liquids (ILs), 1−ethyl−3−methylimidazolium bis (trifluoromethylsulfonyl) imide ([EMIM]+[TFSI]−, cation–anion pairs) inspired by neural mechanoreceptors and engraved electrodes. When different forces (normal, shear, and torsion) were applied to the sensor, the contact area between the ionic gel and the bottom electrode changed through the ionic squeezing effect, and the state of the ionic gel changed; the forces were 0.92 kPa^−1^ (pressure), 5.89 N^−1^ (shear), and 0.12 N cm^−1^ (torsion). A sensor was made by vertically stacking capacitors (Mochtar Chandra et al. [120]). The sensor showed more than quadruple the spatial resolution enhancement from a planar arrangement. As shown in Figure 7d, the structure of the sensor is two capacitors with two capacitors stacked in the *z*-direction, a top electrode, a common electrode, and a bottom electrode, with PDMS as the intermediate medium. This sensor is simple to produce and can detect normal force and shear force and, at the same time, identify the shear angle according to the expression. The expression is as follows:(1)$$\theta =\tan^{-1}\{\frac{2C_{0}\left(C_{A2}-C_{B2}\right)y}{\left(C_{A2}+C_{B2}\right)\left\{2C_{0}-\left(C_{A2}+C_{B2}\right)\right\}x}\}$$
where C_0_, C_A2_, C_B2_, x, and y are the capacitance of C_A_ or C_B_ without force, capacitance of C_A_ with force, capacitance of C_B_ with force, capacitor dimension in the *x*-direction, and capacitor dimension in the *y*-direction, respectively. Ideally, C_0_ = C_A1_ = C_B1_, where C_A1_ and C_B1_ are the capacitances of C_A_ and C_B_ without force (Figure 7d), respectively.

An increase in sensor sensitivity is achieved by reducing the stiffness of the material and structure, as shown in Figure 8a, specifically by reducing the Young’s modulus of the elastomer separating the two electrodes of the capacitor to make it more susceptible to deformation, and the experimental results showed that the sensitivity of the sensor to normal and shear forces increases as expected, while the shear angle detection capability remains unchanged (Yu-Hong Gao et al. [121]). The maximum sensitivity enhancement compared to a typical CTS is 217.6% and 661.3% for normal and shear forces, respectively. The proposed capacitive pressure sensor consists of five electrode cells and was shown to solve the coupling problem within the normal force and shear force by the unique design of the electrode shape (Min-Sheng Suen et al. [122]). Figure 8b shows the specific structure, sensing mechanism, and capacitance variation curve when applying normal, shear, and torsional forces to the sensor. For the first time, using hierarchically engineered elastic carbon nanotube (CNT) fabrics, this sensor can detect pressure and flexion sensing capabilities as well as temperature and humidity. Therefore, it has great potential for e-skin applications (So Young Kim et al. [123]). The sensor is a capacitive pressure sensor consisting of CNT microyarns and Ecoflex dielectric, with PDMS used as an adhesion layer to inhibit the delamination or peeling of the CNT microyarns from the PDMS substrate, as shown in Figure 8c.

In addition to the above [124,125,126,127,128], multiaxis capacitive sensors are also used to detect multiple forces in addition to pressure. For the understanding and comparison of sensors which have several sensing mechanism and detect various types of force, we analyzed and compared stress, strain, stability, and sensitivity of sensors as mentioned above in Table 1. 

## 4. Applications

Pressure sensors can be seen everywhere in daily life, particularly in recent years with the development of flexible pressure sensors with high stability, lightweight, high flexibility, simple production methods, and high sensitivity. The above advantages allow pressure sensors to be used as electronic skin, human–computer interaction interfaces, and intelligent wearable devices.

### 4.1. Electronic Skin (E-Skin)

The skin is the largest organ in the body and can simultaneously sense many stimuli, one of which is pressure [129,130]. Electronic skin has ushered in new developments of technology through pressure sensors to enable robots to have the same sensory capabilities as humans [131,132]. Inspired by the fingertip skin, a highly sensitive pressure sensor with a layered structure of conductive graphite/polydimethylsiloxane foam can be used for electronic skin, and the sensor can sense pressure, texture roughness, and temperature, as shown in Figure 9a (Qi-Jun Sun et al. [133]). The sensor is mounted on a robotic arm, and, depending on the current response obtained from different grasping weights, the robotic arm can obtain “haptics”. Also inspired by human skin, a skin-inspired highly stretchable and conformable matrix network (SCMN) was presented (Qilin Hua et al. [134]). As shown in Figure 9b, experiments have shown that this sensor can be used to identify the location of the pressure load, and the sensor can measure a variety of stimuli in addition to pressure only. A high-sensitivity capacitive pressure sensor was developed with a fingerprint-like pattern embedded in a 3D printed robot arm with a capacitive-piezoelectric tandem sensing structure, and it can detect and discriminate between spatiotemporal tactile stimuli, including static and dynamic pressures and textures (William Navaraj et al. [135]). The sensor can detect and differentiate spatiotemporal tactile stimuli, including static and dynamic pressure and texture, and simulate the real tactile sensation of human hands, shown in Figure 9c. The robotic arm is made “tactile” using an inexpensive randomly structured polystyrene (PS) as the base polymer and using electrostatic spinning to produce a functional fiber pressure sensor (Tamil Selvan Ramadoss et al. [136]). The sensor recognizes the applied pressure using the open-circuit voltage, as shown in Figure 9d, which is mounted on a robot arm and varies over time as the weight is grasped and released. In addition to the commonly used capacitive e-skin, a triboelectric nanogenerator (TENG) with self-powered mechanism was fabricated by easily replicating the surface morphology of natural plants and making interlocking microstructures on the friction layer, as shown in Figure 9e (Guo Yao et al. [137]). The previously mentioned sensors of [96,98,99,123,127] similarly elaborated the applications in electronic skin.

### 4.2. Human–Computer Interaction Interface (HMI)

With the development of society, the HMI is rapidly developing as a platform for information exchange between humans and machines [138,139,140]. Pressure sensors are one of the most basic components of the HMI and are attracting increasing attention for their diverse HMI applications. An electronically tattoo-like frictional self-powered wearable sensor based on vertically aligned vertical gold nanowires (v−AuNW) was reported, where the v−AuNW is sandwiched between two ultrathin layers of spin-coated Ecoflex, resulting in a thin, flexible, and stretchable frictional electric pressure sensor (Tiance An et al. [141]). This was successfully used to control electric lights and toy cars, as shown in Figure 10a. An integrated sensor patch (ISP) was developed, which is insensitive to off-axis deformation and can independently sense tensile strain and external pressure (Hongcheng Xu et al. [142]). In experiments, the authors applied this sensor to a glove to verify its application as a human–machine interface, as shown in Figure 10b, for gesture control. A conductive elastic film with arch-shaped micropatterning arrays (P−HCF) on the surface was fabricated by doping 1D carbon fiber (CF) and 0D carbon nanoparticles (CNP) into PDMS matrix (Mengjuan Zhong et al. [143]). This film was also applied as a human–machine interface on a glove, as shown in Figure 10c, which can be used to simulate hand gestures and control games. A piezoresistive pressure sensor manufactured by combining a laser-induced serpentine graphite structure with an elastomeric substrate (PDMS or Ecoflex) was mounted on a robot arm as an HMI to detect the strain caused by an applied force, as shown in Figure 10d. The sensor can prevent damage to an object by applying excessive force (Yichuan Wu et al. [144]). In addition, the previously mentioned sensors [100,103,119,126,145] were described for HMI applications.

### 4.3. Wearable Devices

The development of smart wearable devices has attracted considerable research attention in recent years [146,147,148], particularly in medicine, mostly used to detect pulse rate, blood pressure, and respiration [149,150,151]. Heart rate testing is an integral part of a wearable device and can detect heart disease, atherosclerosis, cardiomyopathy, and a range of other heart-related conditions [152,153]. Wearable devices highly resistant to interference for accurate real-time detection have attracted considerable scientific interest. Jiang He et al. [154] reported a mass-producible and all-solution processable but straightforward and environmentally friendly strategy for high-performance piezoresistive pressure sensors based on highly conductive interfacial self-assembled graphene (ISG) films. The sensor has high sensitivity and an extensive linear detection range. Figure 11a shows the different pulse data displayed under different motion conditions. The sensor can be used as a high-precision wearable pulse detection system. Ashok Chhetry et al. [155] reported an emerging class of solid polymer electrolytes obtained by incorporating a room-temperature ionic liquid, 1−ethyl−3−methylimidazolium bis(trifluoromethylsulfonyl)imide with poly(vinylidene fluoride-co-hexafluoropropylene) as a high-capacitance dielectric layer for interfacial capacitive pressure sensing applications. A capacitive pressure sensor can detect wrist pulses and blinking movements in wearable devices because of its high sensitivity and low dynamic response time (Figure 11b). Blood pressure is one of the most important parameters of human health. Since the use of blood pressure monitors, the techniques and tools to achieve this detection have not changed significantly, relying mostly on mercury columns and stethoscopes to listen to the Koch sound, which has many limitations. Marian Ion et al. [156] fabricated a piezoresistive pressure sensor using very sensitive microfluidic and biocompatible material. The sensor could be integrated into a wearable device for continuous blood pressure monitoring. Compared with the original blood pressure testing instrument, it is simple to test and very convenient to carry. On the other hand, through experiments, the pulse is measured by placing the sensor at the wrist, as shown in Figure 11c. On the other hand, the current sensor is still relatively resistant to temperature changes, and further improvements are needed. A capacitive pressure sensor with a porous PDMS layer between flexible polyethylene terephthalate (PET) electrodes coated with indium tin oxide (ITO) was proposed as a wearable device for blood pressure detection (Bijender et al. [157]). A flexible pressure sensor based on porous graphene (PG) with polyethylene terephthalate/polyvinyl acetate (PET/EVA) laminate film was proposed to protect the sensor from the skin surface, as shown in Figure 11d. This sensor could be used as a wearable device for blood pressure detection (Yuxin Peng et al. [158]). In addition, the breathable and waterproof nature of the PU film did not affect the sensor significantly when sweating. With the development of technology, the concern of human beings regarding sleep apnea is also increasing. The detection of daily breathing can allow the timely detection of bronchitis, asthma, emphysema, and other respiratory diseases. Respiratory breathing monitoring before and after exercise was reported using a capacitive pressure sensor based on composite nanofiber support, which was mounted on a mask for respiratory testing, as shown in Figure 11e. This sensor exhibited high stability (Sudeep Sharma et al. [159]). On the other hand, more various respiration detection sensors were developed because of the inconvenient nature of the mask. For example, a wireless smart mask for real-time monitoring of breathing conditions in everyday applications was described (Junwen Zhong et al. [160]). The smart mask was designed by integrating an ultrathin self-powered pressure sensor and a compact wireless readout circuit into a normal mask. The pressure sensor consisted mainly of two Au/parylene/Teflon AF films (micron thickness) and was electrostatically self-powered. The ultrathin wireless transmission makes it very easy to use, as shown in Figure 11f.

## 5. Conclusions and Prospects

This review presented a recent comprehensive advance in the development of sensors that can detect single forces and multiforces through the novel design of device structures. A combination of new materials and novel structural designs achieved outstanding sensing capabilities in response to multiple external stimuli, all of which will lead to applications in wearable devices, HMI, electronic skin, automatic industries, and automatic driving. Various sensing mechanisms (piezoresistive, piezoelectric, and capacitance) have advantages in the sensing capabilities and fabrication process. Therefore, there are many studies for fabricating sensors that can detect various physical quantities, but challenges remain for practical applications, such as sensors. Further innovations in materials and device structures are needed to sense various physical quantities for multiple-sensor sensing. For detecting various physical quantities, the novel structural design in response to multiple external stimuli is essential for discriminating various external stimuli. The development of electric signal analysis is also required to distinguish various physical quantities. For the accurate discrimination of external forces and analysis of electric signals from physiological signals, signal processing technology should be developed for collecting, digitizing, and analyzing electric signals. These innovations in materials, novel structures, and analyzing technology will enable further improvement of sensors that will apply to the automatic systems and metaverse to come in the near future.

## Figures and Tables

**Figure 1 sensors-23-04300-f001:**
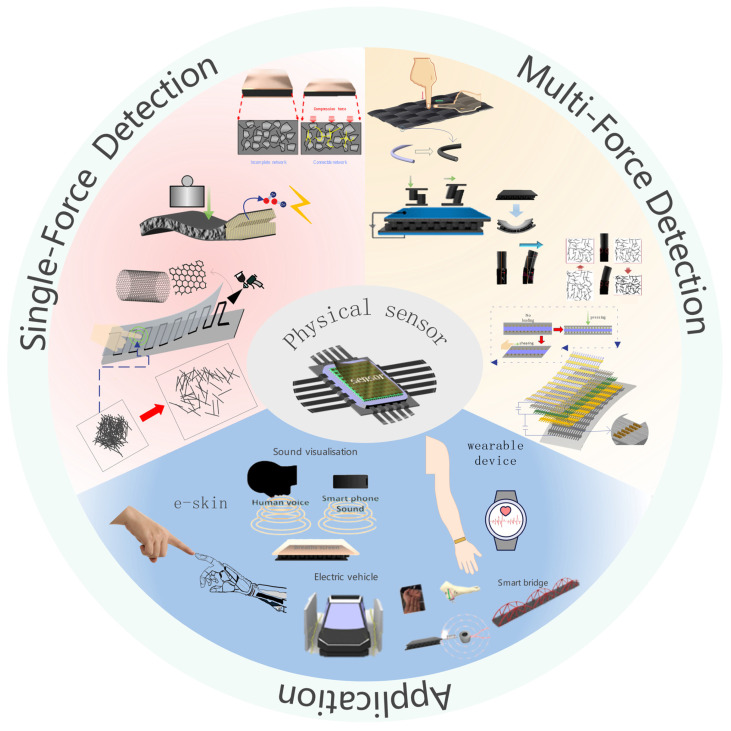
Schematic showing single and multiforce detection and application of the physical sensor.

**Figure 3 sensors-23-04300-f003:**
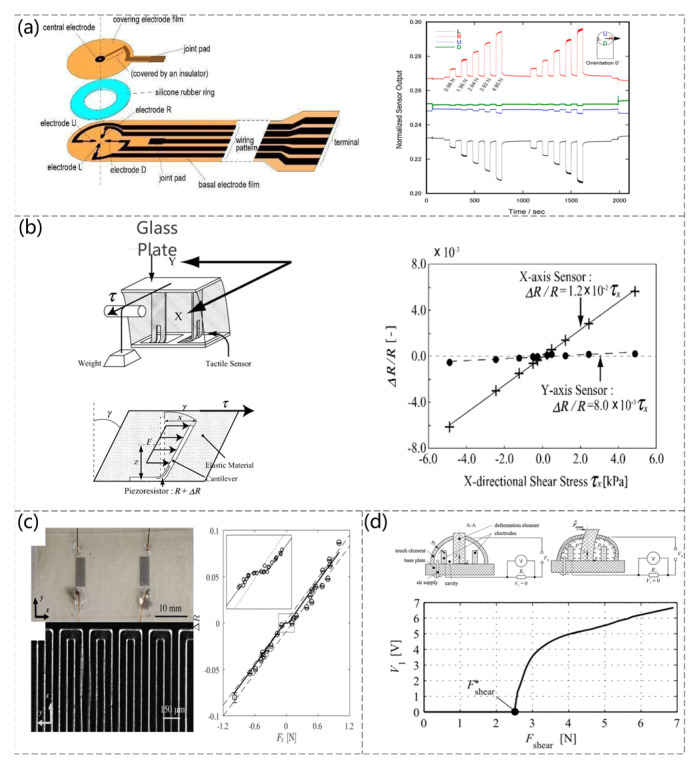
(**a**) Schematic diagram of the sensor structure and the time course of the normalized sensor output during the application of shear force. Reproduced with permission from Sensors (2017) [69]. (**b**) The working mechanism of the sensor and the change in resistance of sensors when shear stress is applied. Reproduced with permission from Sensors and Actuators A:Physical (2006) [72]. (**c**) Microscopic view of an assembled shear sensor skin showing a pair of eGaIn microfluidic channels embedded in PDMS and serpentine microchannels adjacent to the reservoir and experimental circles and computational (solid line) normalized resistance sensor response as a function of the applied shear force. Reproduced with permission from Sensors and Actuators A:Physical (2017) [73]. (**d**) Structural working principle of the sensor and dependence of the voltage V1 on the applied shear force at an internal pressure of *p* = 0 MPa. Reproduced with permission from Sensors and Actuators A:Physical (2016) [74].

**Figure 4 sensors-23-04300-f004:**
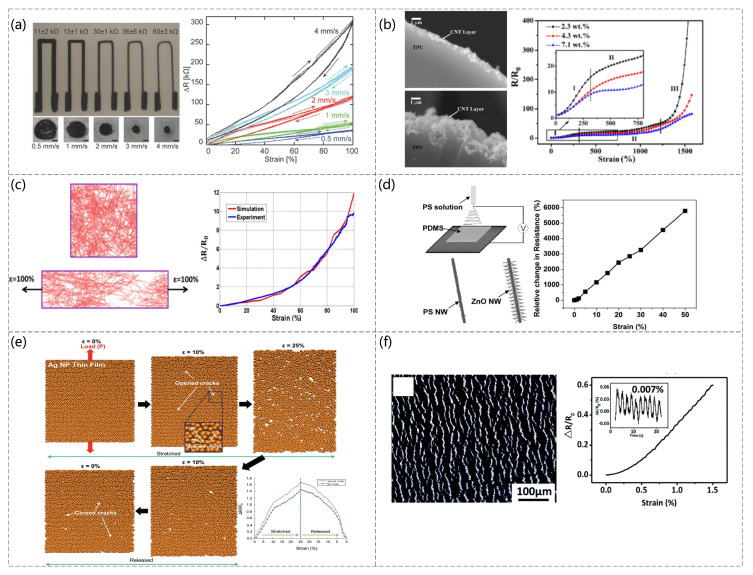
(**a**) e−3DP−printed images of the top and cross-section of a soft sensor at different speeds, and images of the change in deformation rate vs. resistance value at different speeds. Reproduced with permission from Advanced Materials (2014) [76]. (**b**) The 2.3 wt.% and 10 wt.% CNT loading and relative resistance of CNTs−TPU fibers with various contents of CNTs under stretching at a displacement rate of 5 mm/min (sample length is 20 mm); the inset shows a magnified image of the relative resistance change in the strain range from 0% to 750%. Reproduced with permission from Carbon (2012) [77]. (**c**) Sensor principles and the response of AgNWs−PDMS nanocomposites applied strain through experimental measurements and numerical simulations. Reproduced with permission from ACS (2014) [78]. (**d**) Schematic diagram of the fabrication of a hybrid structure of ZnO NWs/PSNFs and the relative change in resistance of the sensor at different strains. Reproduced with permission from Advanced Materials (2011) [79]. (**e**) Numerical simulation results based on molecular dynamics of Ag NP films during elongation/relaxation and the relative change in resistance of Ag NP films with and without initial microcracking during stretching/release, as calculated by a numerical simulation. Reproduced with permission from Nanoscale (2014) [80]. (**f**) SEM image of the 10 layers of CNTs films and relationship between the strain pressure and resistance change after the composition of the sensor. Reproduced with permission from Journal of Materials Chemistry C 2018 [81].

**Figure 5 sensors-23-04300-f005:**
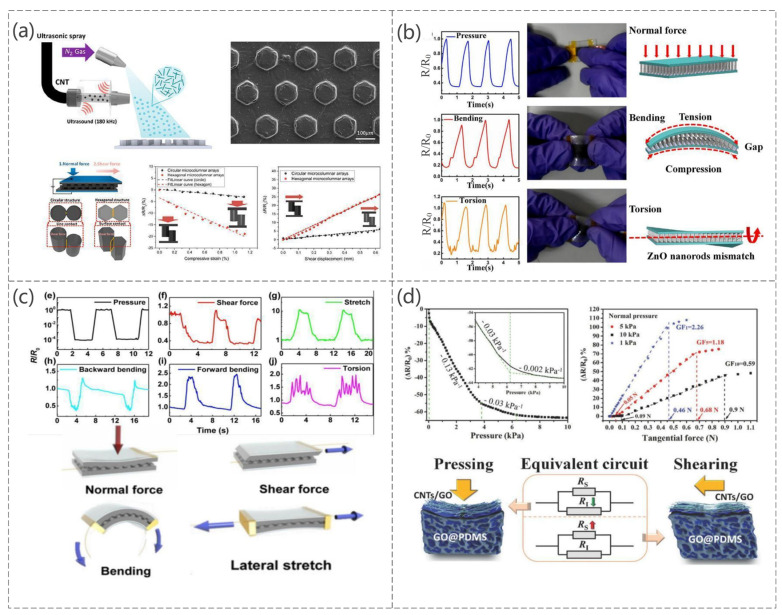
(**a**) The process of making this sensor, SEM images, and the relationship between the force applied and the relative resistance under different force conditions. Reproduced with permission from ACS (2020) [96]. (**b**) Change in the relative resistance to distinguish different types of mechanical pressure pressing, bending, and torsion. Reproduced with permission from Sens Actuators A (2018) [97]. (**c**) Simulation of the relative resistance change in the electronic skin in response to different mechanical stimuli and the deformation that occurs when subjected to different mechanical stimuli. Reproduced with permission from ACS (2014) [98]. (**d**) Variation of the relative resistance of the electronic skin as a function of normal compression, variation of the relative resistance with tangential force at 5 kPa or 10 kPa normal pressure, schematic representation of the deformation of the shape of the contact point under normal pressure and lateral shear, and the corresponding variation of the resistance in the equivalent circuit of the electronic skin. Reproduced with permission from Adv funct Mater (2018) [99].

**Figure 6 sensors-23-04300-f006:**
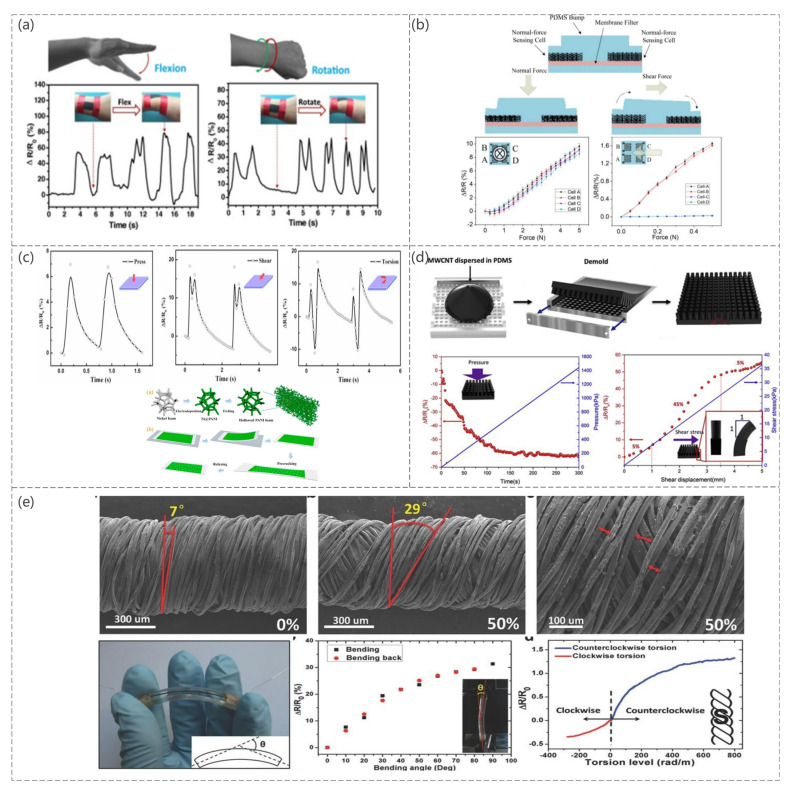
(**a**) Monitoring of the wrist motions using FSSF: wrist flexion and rotation. Reproduced with permission from Chem Eng J (2017) [100]. (**b**) Schematic diagram of the sensor operation and the relative resistance change curve under different mechanical stimuli. Reproduced with permission from Sens Actuators A (2015) [101]. (**c**) Illustration of the fabrication process of a freestanding PANI foam. PF/PDMS and C−PF/PDMS composites and pressure sensor subjected to compression, shear, and torsion. Reproduced with permission from Composites Part A (2019) [102]. (**d**) Schematic diagram of the sensor fabrication and the relative resistance changes under different mechanical stimuli. Reproduced with permission from Composites Science and Technology (2019) [103]. (**e**) SEM image of PDCY−RGO strain. Photograph of the bending sensor. Inset: The definition of bending angle θ in a schematic structure, resistance variation of the bending sensor in forward and reverse directions. Inset: Bending sensor in testing and resistance variation of the graphene-based fiber in torsion test from 280 rad m^−1^ to 800 rad m^−1^. Inset: Structure drawing of the twisting PE fiber (outer layer of S-twist). Reproduced with permission from Adv Mater (2015) [104].

**Figure 7 sensors-23-04300-f007:**
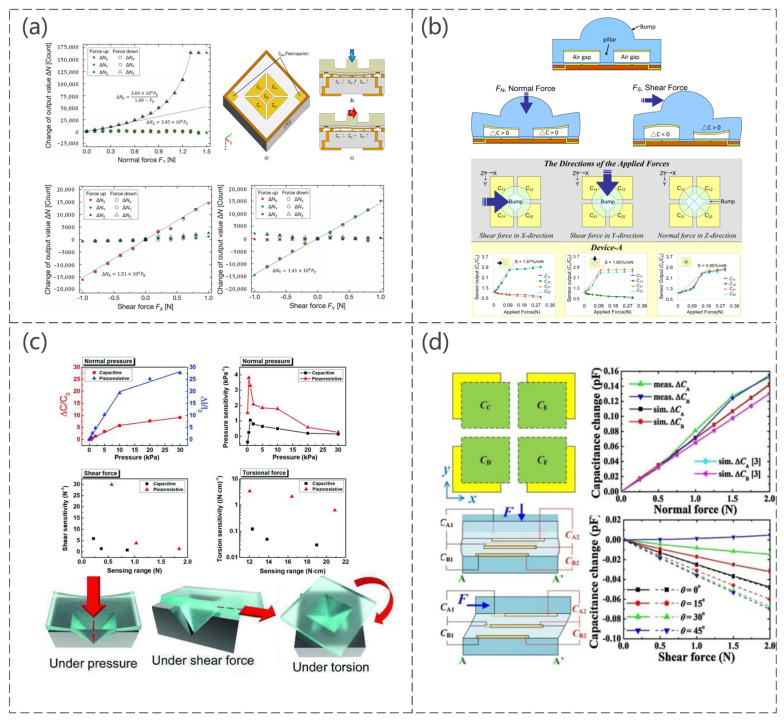
(**a**) Electrode layout; working principles for normal force FZ and X-axis shear force FX, change in the output value of the three-axis integrated tactile sensor by normal force FZ; shear force FX and shear force FY. Reproduced with permission from Sensors (2017) [117]. (**b**) Schematic diagram of a shear stress sensing element without applied forces, with a normal force, a shear force, and measurement of capacitance in relation to an applied force (N). Reproduced with permission from Sensors (2010) [118]. (**c**) Capacitance/resistance curve for different mechanical stimuli and schematic diagram of sensor operation. Reproduced with permission from Adv Mater Technol (2019) [119]. (**d**) Structure and operating principle of the sensor, the capacitance change curve when subjected to pressure, and the simulated response pattern of a shear force at different plane shear angles. Reproduced with permission from Sens Actuators A (2017) [120].

**Figure 8 sensors-23-04300-f008:**
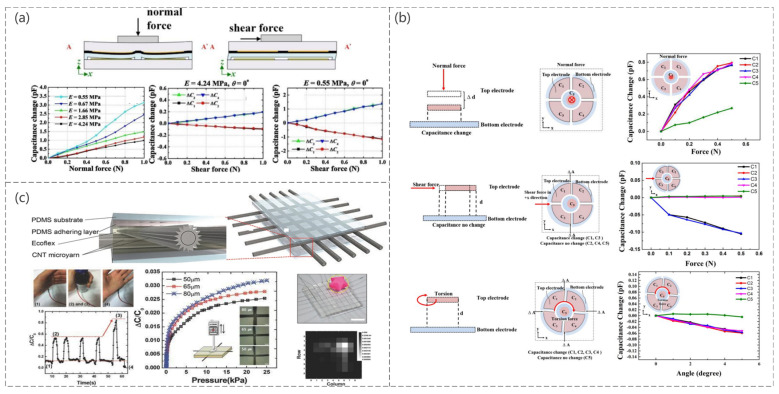
(**a**) Schematic diagram of the sensor operation and the capacitance change curve when subjected to normal and shear forces. Reproduced with permission from Sens Actuators A (2019) [121]. (**b**) Schematic diagram of the sensing mechanism in a normal force with the measured capacitance changes for the normal force in the –*z*-direction, schematic diagram of the sensing mechanism in shear force with the measured capacitance changes for the shear force in the *x*-direction, and schematic diagram of the sensing mechanism in torsion with the measured capacitance changes for the torsion in the x−y plane. Reproduced with permission from Sens Actuators A (2018) [122]. (**c**) Schematic diagram of the structure of the sensor, the corresponding curve when mounted on the finger when bent, the corresponding curve when subjected to pressure, and the spatial distribution of the object on the sensor array. Reproduced with permission from Advanced Materials (2015) [123].

**Figure 9 sensors-23-04300-f009:**
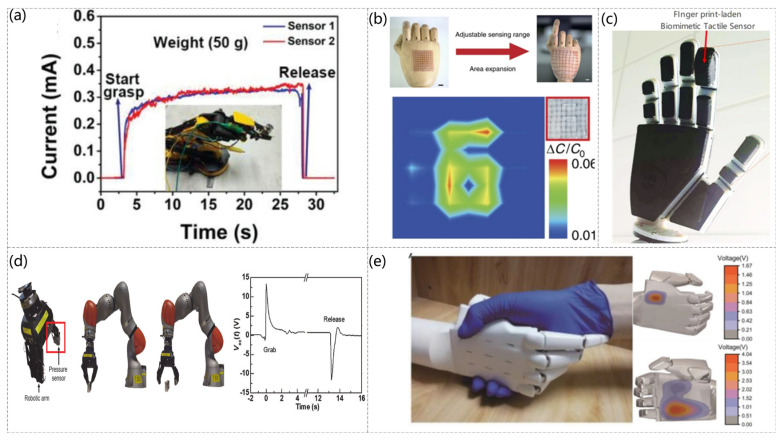
(**a**) Photograph of two skin sensors attached to a robotic arm. Current responses of the skin sensors to the applied pressure in grasping and releasing a weight of ≈50 g. Reproduced with permission from Advanced Functional Materials (2019) [133]. (**b**) Schematic diagram of the SCMN as an artificial electronic skin on the hand, showing the adjustability and scalability of the sensing (scale bar: 1 cm) and the pressure mapping of the pressure sensor array with the number “6” printed on it at a pressure of 8 kPa. Reproduced with permission from Nat Commun (2018) [134]. (**c**) 3D-printed hand with a fingerprint-laden biomimetic sensor stack. Reproduced with permission from Advanced Intelligent Systems (2019) [135]. (**d**) Pressure sensor attached to a robotic arm-robotic arm grab and release mechanism. Open-circuit voltage with time when the robotic arm grabbed and released the weight. Reproduced with permission from Nanomaterials (2021) [136]. (**e**) Photograph of human-robot handshaking and the voltage contour profiles on the back and palm of the bionic hand during handshaking. Reproduced with permission from Advanced Intelligent Systems (2020) [137].

**Figure 10 sensors-23-04300-f010:**
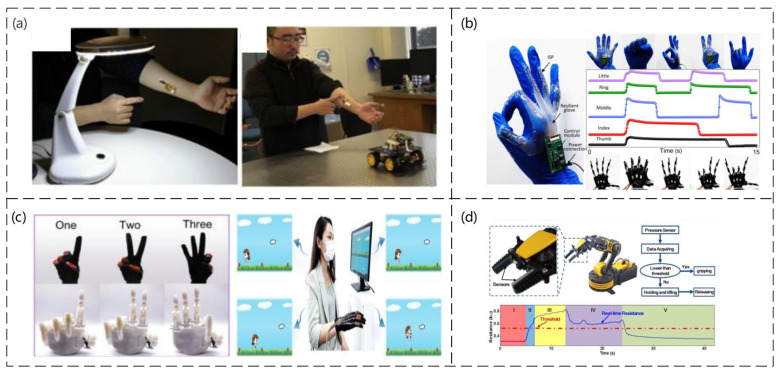
(**a**) HMI for wireless light switches and HMI for wireless vehicle controllers. Reproduced with permission from Nano Energy (2020) [141]. (**b**) Optical image of the entire wearable stretchable glove connected to the five ISPs and the processing circuit. Images of hand gestures and their corresponding machine hand gestures, the output digital signal of the processing circuit with different gestures. Reproduced with permission from Nano Energy (2022) [142]. (**c**) Imitated gestures and the smart glove used in intelligent rehabilitation training by playing PC games. Reproduced with permission from Chemical Engineering Journal (2021) [143]. (**d**) An application demonstration of the strain sensor in a robotic hand system. Reproduced with permission from Sensors and Actuators A: Physical (2018) [144].

**Figure 11 sensors-23-04300-f011:**
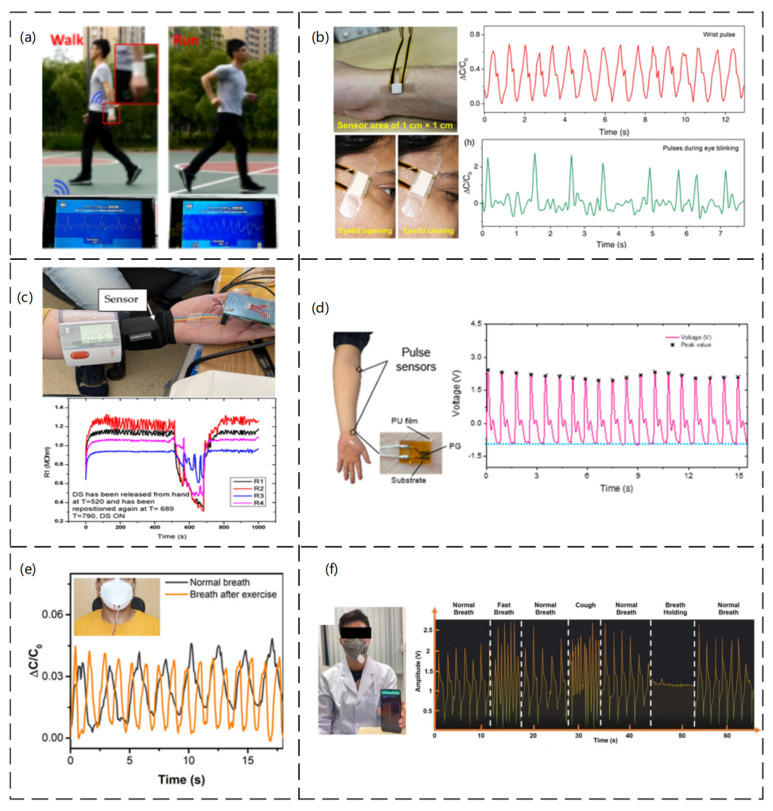
(**a**) Image of wireless health monitoring system applied while walking and running. Reproduced with permission from Nano Energy (2019) [154]. (**b**) Real-time monitoring of the capacitive response of the wrist pulse sensor and the photographic corresponding blinking action of a sensor used to detect muscle movements during eyelid opening and closing. Reproduced with permission from ACS (2019) [155]. (**c**) Measurement setup (DC−2) for evaluating the FPS. The sensor is placed on a radial artery, covered by a Velcro band, and placed immediately next to it is the digital sphygmomanometer; graphical representation of the measuring setup (DC−2) for an evaluation of the FPS. Reproduced with permission from Sensors (2021) [156]. (**d**) Sensor placement and corresponding curves. Reproduced with permission from Sensors (2021) [158]. (**e**) Respiration monitoring under pre-and post-exercise conditions. Reproduced with permission from ACS (2020) [159]. (**f**) Photograph illustrating the wireless breath-monitoring process. Photographs showing the measured continuous and successive breath conditions of normal breath, fast breath, normal breath, cough, normal breath, breath holding, and normal breath of the tester. Reproduced with permission from Advanced Materials (2022) [160].

**Table 1 sensors-23-04300-t001:** Comparison of stress, strain, stability, and sensitivity of sensor.

Stress	Strain	Stability	Sensitivity	Reference
Pressure 0–2.5 kPa (M15/CM250 3 wt% 2:8)	0–1%	/	Gauge factor 25 M15/CM250 3wt% 7:3	[61]
Pressure(0–500 kPa)	/	1000 [cycles]	7.62 kPa^−1^	[62]
Pressure(0–35 kPa)	/	2000 [cycles]	56.15 kPa^−1^	[64]
Pressure (0–5 kPa)	/	10^5^ [cycles]	(60 V, <5 kPa) 192 kPa^−1^ (6–24 V) 0.23–14.2 kPa^−1^	[65]
Pressure(0–400 kPa)	/	10^4^ [cycles]	0.18 kPa^−1^	[67]
Normal force (0.5~5 N)	/	10 [cycles]	540 mV/N	[68]
Shear force (0~5 N)	/	/	/	[69]
Shear force (−5.0~5.0 kPa)	/	/	Resistance change: Vertical direction 1.30 × 10^−3^ kPa^−1^ Parallel direction 0.06 × 10^−3^ kPa^−1^	[72]
Shear force (−2~2 N)	/	/	0.078 N^−1^	[73]
Shear force (0–7 N)	/	/	/	[74]
/	0~400%	1000 [cycles]	/	[76]
/	0~1500%	800 [cycles]	/	[77]
/	0~70%	1000 [cycles]	Gauge factor ∼5	[78]
/	0~50%	/	/	[79]
/	0~20%	1000 [cycles]	0.35 kPa^−1^	[80]
/	0~100%	1500 [cycles]	Gauge factor 87	[81]
Pressure (0–22 kPa), shear displacement (0–1 mm), bending curvature (0–0.18 mm^−1^)	/	1000 [cycles]	1.39 [mm^−1^]	[96]
Pressure (0–14 kPa), bending, torsion	/	/	ZnO: −0.768 kPa^−1^ AZO: −0.223 kPa^−1^	[97]
Pressure (0–25 kPa), shear force (0–6 kPa), bending, Stretching, twisting	ΔL/L0 (0–1.2) 0~120%	1000 [cycles]	2.21 N^−1^ at 65 Pa	[98]
Pressure (0–10 kPa), tangential force (0–1.1 N), bending angle (0–180°)	0–45%	5000 [cycles]	−0.31 kPa^−1^ at 0.05–3.8 kPa, −0.03 kPa^−1^ at 3.8–6.3 kPa.	[99]
Bending, torsion	0–30%	8 [washing cycles]	/	[100]
Normal force (0–5 N), shear force (0–0.5 N)	/	/	Part A (2.1086%/N) Part B (1.952%/N) Part C (2.053%/N) Part D (1.828%/N)	[101]
Pressure (0–5 kPa), shear, torsion	0–23%	500 [cycles]	0.055 kPa^−1^ at 0–5.0 kPa Strain GF(10 (R 2 > 0.998))	[102]
Pressure (~1200 kPa) shear (0~5 mm)	/	/	Pressure 0.075 kPa^−1^ Shear 2 kPa^−1^	[103]
Bending angle (0–90°), torsion level (−280 rad m^−1^~800 rad m^−1^)	0–100%	10,000 [cycles]	Detection limit of 0.2% strain	[104]
Normal force (0–1.5 N), shear force (–1~1 N)	/	/	Pressure 34 count/mN Shear 14 to 15 count/mN	[117]
Normal force (0–0.36 N) shear force (0–0.36 N)	/	/	1.67%/mN	[118]
Pressure (0–50 kPa), shear force (0–1 N), torque (0–20 N cm)	/	/	Capacitive: 0.92 [kPa^−1^] (<10 kPa) 0.10 [kPa^−1^] (10–50 kPa) Piezoresistive: 1.93 [kPa^−1^] (<10 kPa) 0.42 [kPa^−1^] (10–30 kPa) 5.89 N^−1^ (shear force) 0.12 (N cm)^−1^ (torsion)	[119]
Normal force (0–2 N), shear force (0–2 N)	/	/	(71 fF/N) normal force sensitivity (37 fF/N) shear force sensitivity at θ = 45°	[120]
Normal force (0–1 N), shear force (0–1 N)	/	/	F = 1 N E = 0.55 MPa sensitivity 3.14 pF/N	[121]
Normal force (0–0.5 N), shear force (0–0.5 N), torsion angle(0–5 degree)	/	/	Normal sensitivity 1.558 pF/N Shear sensitivity −0.833 pF/N Torsion sensitivity −0.094 pF/N	[122]
Pressure (0–25 kPa), bending	/	2000 [cycles]	80 μm: Low pressure −0.05 ± 0.01 [kPa^−1^] High pressure −0.6 [kPa^−1^]	[123]

## Data Availability

Not applicable.

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
