# Peer review of "Recent Development of Mechanical Stimuli Detectable Sensors, Their Future, and Challenges: A Review"

_sensors, 2023, doi:10.3390/s23094300_

Round 1

Reviewer 1 Report

There is lot of work is produced by thr researchers on mechanical stimuli and detections, authros have tried to combine all the efforts in a single article. Howver there are few things to be considered and resubmit it after revising.

1. abstract is not well written and distinguished. Authors need to provide some specific results and strategies carried out by the others.

2. Introduction is ver poorly written, it needs to be re-written again. there are issues in english language as well througout the manuscript, so it needs to those needs to be addreesed.

3. I would like to see the comparsion of this resdaerch in terms of well defined tables for stress, strain, stability, sensitivity etc

4. Some recent articles should be added and cited 

Reviewer 2 Report

In this paper, the authors try to write a review paper for recent development of mechanical stimuli detectable sensor, their future and challenges. Unfortunately, the paper consist of only a stack of many reference citations and lack the indepth of analysis, which, this reviewer believe, is the core of a review paper.   

Reviewer 3 Report

The paper is devoted to illustrating many studies about sensors detecting various forces. The authors classify sensors that acquire one or many mechanical variables. They also stress the possible applications of the sensors considered. 

The reviewer finds no fault whatsoever with the selected sensors, the discussions, and the conclusions. It is an interesting contribution, and hence he recommends the publication after minor revisions.

Please, see the question below:

1) The writing, while generally acceptable for non-native English speakers, is a bit choppy. I suggest that the authors carefully proofread their paper.

2) After Fig. 5, it is impossible to distinguish where the caption ends and where the text of the paper starts. Please check and correct.

3) On page 10, line 266, the sentence "as shown in Figure (6a)." appears to be fragmented. 

4) In the text when the authors talk about artificial skin, the reviewer suggests also citing [1] where wearable sensors are considered and [2] where it is possible to find a model for these wearable sensors attached to the skin.

[1] D.H. Kim, N. Lu, R. Ma, Y.S. Kim, R.H. Kim, S. Wang, J. Wu, S.M. Won, H. Tao, A. Islam, et al., Epidermal electronics, Science 333 (2011) 838–843.

[2] Giorgio, I., Della Corte, A., dell'Isola, F., & Steigmann, D. J. (2016). Buckling modes in pantographic lattices. Comptes rendus Mecanique, 344(7), 487-501.

Round 2

Reviewer 1 Report

Tha data provided in the article is enough to be published. There are follwoing few points can be kept in mind while revising the manuscript.

1. Kindly check the grammer throughout the masucript

2. revise conclusion part